# Unveiling And Addressing Dimensional Collapse In Vector Quantization Models Via Codebook Regularization

Fang Zhang [* 1 2]   Yongxin Zhu [* 1]   Yihao Liu [2]   Bin Fu [2]   Linli Xu [1]

## Abstract

While recent advancements in Vector Quantization (VQ) models have successfully achieved complete codebook utilization, a critical bottleneck remains largely unexplored: the effective dimensionality of the codebook embedding space. We observe that discrete codebook representations tend to degenerate into low-dimensional subspaces, characterized by significantly lower effective rank than continuous representations during quantization. Through comprehensive spectral analysis, we identify that this dimensional collapse stems from the suppression of low-variance components inherent to the vector quantization process, thereby severely limiting the expressive capacity of VQ models. To mitigate this fundamental issue, we propose a simple yet effective codebook regularization strategy designed to restore low-variance components, effectively bridging the spectral gap between discrete codebook spaces and continuous representations. Extensive experiments demonstrate that this regularization objective is compatible with diverse VQ training paradigms, yielding significant improvements in reconstruction fidelity and downstream performance in autoregressive image generative models. Code is available at `https://github.com/ksblk2116/dimvq`.

## 1. Introduction

Vector Quantization (VQ) (van den Oord et al., 2017; Razavi et al., 2019) has established itself as the foundational paradigm for discrete tokenization, enabling breakthroughs in image (Bao et al., 2022; Esser et al., 2021), video (Wang et al., 2024; Lu et al., 2025; Zhao et al., 2024), and audio synthesis (Baevski et al., 2020; Borsos et al., 2023; Ji et al., 2024). By compressing continuous signals into discrete indices, VQ tokenizers facilitate both representation learning (Baevski et al., 2020; Bruce et al., 2024) and autoregressive generative modeling (Sun et al., 2024; Tian et al., 2024b). While prior methods (Zhu et al., 2024; Yu et al., 2022) struggled with codebook collapse—where only a subset of codes is utilized—recent advancements such as IBQ (Shi et al., 2025) and SimVQ (Zhu et al., 2025) have successfully achieved near-complete codebook utilization via codebook re-parameterization. However, we argue that maximizing codebook utilization addresses only half of the problem. Although current methods ensure complete code usage, they overlook a fundamental quantization bottleneck: the effective dimensionality of the codebook embedding space.

Our investigation reveals a critical phenomenon termed *dimensional collapse*, which occurs when mapping continuous features to discrete codebook spaces through vector quantization. As illustrated in Figure 1 (a), even when the codebook size scales to 262,144 and the utilization reaches 100%, the codebook with high dimension (e.g., 128) exhibits a surprisingly low effective rank—a metric quantifying the number of dimensions actively utilized—compared to the continuous representation of its Autoencoder (AE) counterpart[1] (12.1 vs 37.2). This gap indicates that the codebook space degenerates into a redundant, low-dimensional subspace, severely limiting its representational capacity and degrading reconstruction quality (rFID 5.87 vs 0.27).

To elucidate the underlying causes of dimensional collapse, we first analyze the spectrum of the codebook space and continuous latent space through Singular Value Decomposition (SVD), as shown in Figure 1 (b). A distinct contrast is observed: whereas the continuous AE exhibits a smooth spectral decay, the standard VQ codebook suffers from *spectral truncation*, where singular values corresponding to the low-variance tail drop abruptly to zero. This phenomenon

---

[*]Equal contribution

Work done during Fang Zhang's internship at Shanghai Artificial Intelligence Laboratory. [1]School of Computer Science and Technology, University of Science and Technology of China, Hefei, China [2]Shanghai Artificial Intelligence Laboratory, Shanghai, China. Correspondence to: Fang Zhang <fangzhang@mail.ustc.edu.cn>.

*Proceedings of the $43^{rd}$ International Conference on Machine Learning*, Seoul, South Korea. PMLR 306, 2026. Copyright 2026 by the author(s).

---

[1]This AE is trained without the quantization layer, while the remaining part is consistent with the VQ model in all other aspects.

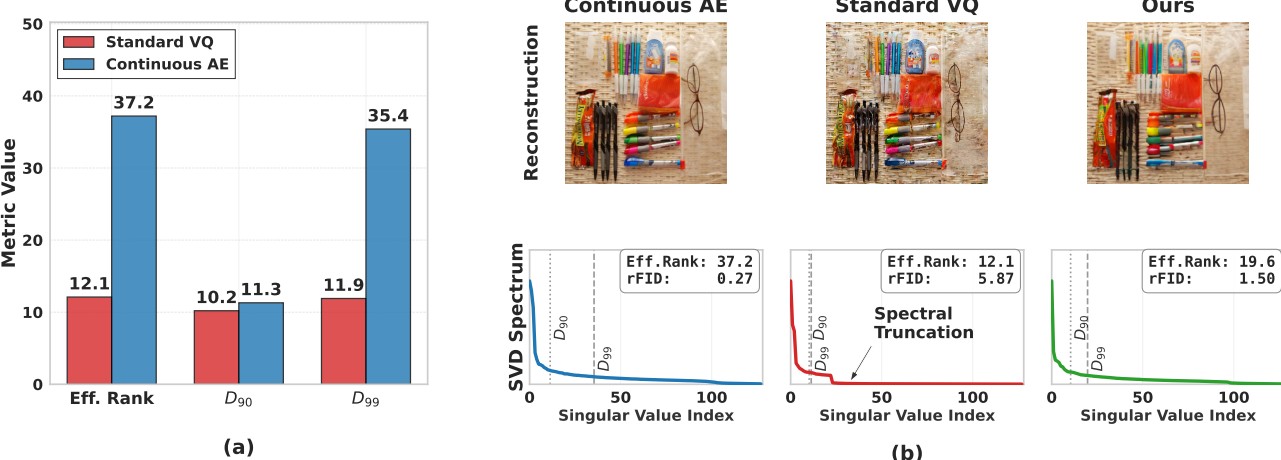

*Figure 1.* (a) Comparison of effective dimensionality between Standard VQ and its continuous AE counterpart. (b) Spectral Truncation and Visual Fidelity: We compare reconstruction performance from a continuous Autoencoder (AE), a standard VQ-GAN, and our enhanced VQ with codebook regularization. Below each image are the SVD spectrum of its codebook/latent feature, along with its effective rank and rFID.

reveals that the dimensional collapse fundamentally stems from the suppression of low-variance components during quantization. This insight is further corroborated by quantitatively decomposing the spectral energy using metrics $D_{90}$ and $D_{99}$ (defined as the minimum dimensions needed to capture 90% and 99% of the total energy, respectively). A comparative analysis shows that while dominant components remain consistent between discrete and continuous representations (similar $D_{90}$ values), a significant discrepancy emerges at the $D_{99}$ level. This confirms that the low-variance components vanish during quantization, resulting in degraded reconstruction fidelity.

Consequently, the key to resolving dimensional collapse lies in mitigating the suppression of these low-variance components to minimize the spectral gap between discrete and continuous representations. To achieve this, a representative approach is Spectrum Hinge, which forces all singular values to exceed a pre-defined threshold. However, applying such hard constraints raises critical issues, including high sensitivity to hyperparameter tuning and aggressive disruption of the data's natural spectral distribution.

To overcome these limitations, we propose a simple yet effective codebook regularization. Instead of rigid thresholding, our approach adopts a redundancy reduction strategy that encourages distinct dimensions to encode diverse information. This mechanism naturally restores the suppressed low-variance components—as demonstrated in Figure 1 (b)—while aligning with the spectral distribution of continuous representations.

Extensive experiments validate that our codebook regularization is compatible with diverse VQ frameworks and substantially improves their representation capacity. For example,

it elevates the IBQ baseline to state-of-the-art reconstruction performance (rFID: 0.91 on ImageNet). Moreover, we demonstrate that tokenizers with high effective dimensionality directly contribute to better generation performance when utilized in downstream autoregressive generation models.

In summary, our main contributions are as follows:

- We provide a spectral perspective on dimensional collapse in the codebook space, identifying that vector quantization inherently suppresses low-variance components, severely limiting its representation capacity.

- We introduce codebook regularization, a simple yet effective strategy to revitalize these dormant low-variance components. By implicitly reducing redundancy, our method aligns the spectrum of discrete codebook spaces with continuous representations, thereby increasing the effective rank.

- We verify the versatility of our regularization by integrating it into multiple VQ-GAN training frameworks, achieving consistent performance gains in both reconstruction and generation benchmarks.

## 2. Background

### 2.1. Vector Quantization

Vector Quantization (VQ) imposes a discrete bottleneck that maps continuous embeddings onto a finite set of tokens. The framework comprises an encoder $\mathcal{E}$, a decoder $\mathcal{D}$, and a learnable codebook $\mathcal{C} = \{e_k\}_{k=1}^{K} \subset \mathbb{R}^{K \times D}$, where $K$ denotes the vocabulary size and $D$ represents the embed-

ding dimension. Given an input $x$, the encoder produces a continuous latent representation $z_e = \mathcal{E}(x) \in \mathbb{R}^d$, which is subsequently discretized to its nearest neighbor $z_q$ within the codebook:

$$z_q = e_k, \quad \text{where } k = \arg\min_j |z_e - e_j|_2^2. \quad (1)$$

To facilitate end-to-end optimization despite the non-differentiable quantization step, the Straight-Through Estimator (STE) is utilized, approximating the gradient as $\partial\mathcal{L}/\partial z_e \approx \partial\mathcal{L}/\partial z_q$ during backpropagation.

The total objective combines a reconstruction term (often enhanced by perceptual loss $\mathcal{L}_P$ and adversarial loss $\mathcal{L}_{Adv}$) with a regularization term. Moreover, to stabilize the discrete mapping, a commitment loss $\mathcal{L}_{commit}$ is introduced to constrain the encoder outputs to commit to the chosen codebook vectors.:

$$\mathcal{L}_{commit} = \|z_e - \text{sg}[z_q]\|_2^2 + \beta\|\text{sg}[z_e] - z_q\|_2^2, \quad (2)$$

where $\text{sg}[\cdot]$ denotes the stop-gradient operator.

### 2.2. Quantitative Metrics: Effective Rank and Spectral Truncation Gap

To quantitatively evaluate the severity of dimensional collapse, we introduce effective rank (ER), which measures the effective dimensionality of the feature space by analyzing the spectral distribution of the embeddings. Following Roy & Vetterli, for a codebook matrix $\mathcal{C}$ with singular values $\{\sigma_k\}$, we first normalize them to obtain a probability distribution $p_k = \frac{\sigma_k}{\sum_i \sigma_i}$. The effective rank is then defined as the exponential of the spectral entropy:

$$\text{ER} = \exp\left(-\sum_k p_k \log p_k\right). \quad (3)$$

A low ER indicates that the codebook has collapsed into a low-dimensional subspace, utilizing only a fraction of its nominal dimension $D$.

To further investigate the phenomenon of spectral truncation, we introduce $D_\alpha$ as the minimum number of principal components required to explain $\alpha\%$ of the total variance:

$$D_\alpha = \min\left\{k \,\middle|\, \frac{\sum_{i=1}^k \sigma_i^2}{\sum_{j=1}^D \sigma_j^2} \geq \frac{\alpha}{100}\right\} \quad (4)$$

where $\{\sigma_i\}$ denote the singular values of the feature embeddings sorted in descending order. In the following experiments, we introduce $D_{90}$ and $D_{99}$ to denote the number of dimensions required to capture 90% and 99% of the total feature space information, respectively.

Intuitively, if the difference between these two metrics is negligible, it indicates a sharp spectral truncation where low-variance components are suppressed.

## 3. Spectral Truncation In Vector Quantization Models

In this section, we investigate whether the phenomenon of spectral truncation is merely an artifact of specific hyper-parameter choices or an inherent property of vector quantization models. Furthermore, we demonstrate that the suppressed low-variance components are critical for preserving fine-grained details.

### 3.1. AutoEncoder as Continuous Baseline

To verify that spectral truncation is a direct consequence of vector quantization, we employ a continuous Autoencoder (AE) as our reference baseline. The AE mirrors the VQ model's architecture and training objectives (including perceptual and adversarial losses), with the only exception that the quantization layer is replaced by an identity mapping. This design ensures that the AE's latent space captures continuous features without the information loss inherent to discretization, serving as the ideal continuous distribution that the discrete codebook space strives to approximate Consequently, any spectral discrepancy between the VQ model and the AE can be unambiguously attributed to the vector quantization process.

### 3.2. Generality of Spectral Truncation

*Table 1.* Default Baseline Configuration. Unless otherwise specified, all ablation studies and comparisons are conducted based on this standard setup.

| Hyperparameter | Default Value |
|---|---|
| Dataset | ImageNet-1K $128 \times 128$ |
| Backbone Architecture | CNN-based |
| Embedding Dimension ($D$) | 64 |
| Codebook Size ($K$) | 1024 |
| Training Paradigm | VQGAN without EMA |

Building on this comparative framework, we conduct a systematic evaluation across a diverse set of VQ configurations to verify the universality of spectral truncation. We anchor our analysis on a standard baseline representing a typical VQGAN setup, as detailed in Table 1.

In subsequent experiments, we systematically vary specific components—including the datasets () backbone (ViT vs. CNN), embedding dimensions ($D \in \{64, 256, 512\}$), codebook sizes ($K \in \{1024, 16384\}$), and training paradigms (EMA)—while keeping other hyperparameters fixed to these baseline values.

As quantitatively illustrated in Figure 2, the AE consistently exhibits a significantly higher effective rank compared to its VQ counterparts. This sharp contrast provides empirical evidence of dimensional collapse, where the codebook manifold degenerates into a redundant, low-dimensional

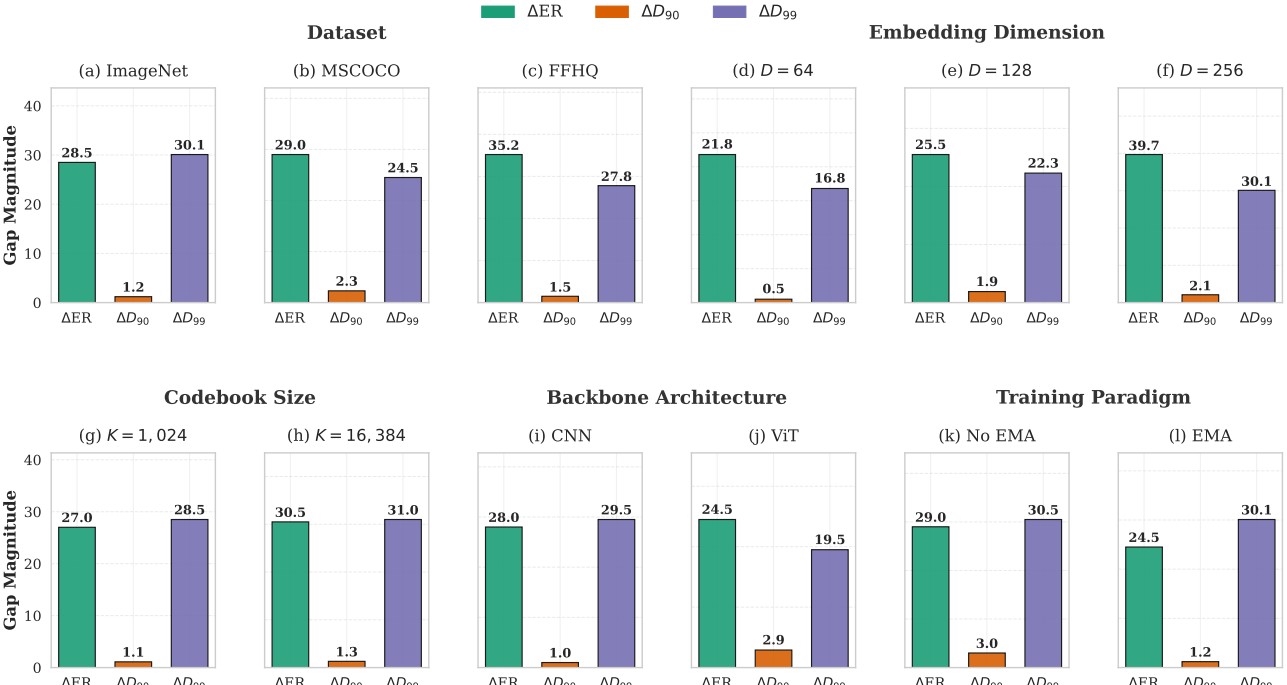

*Figure 2.* We report the quantitative gaps between VQ models and their continuous AE counterparts across diverse experimental settings. The metrics include the difference in Effective Rank ($\Delta$ER), and the difference in the number of dimensions required to capture 90% ($\Delta D_{90}$) and 99% ($\Delta D_{99}$) of the total information.

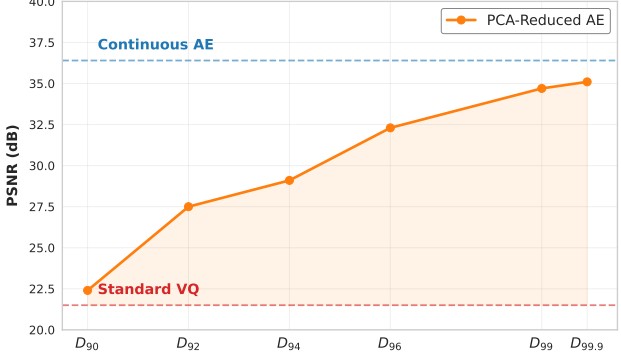

*Figure 3.* The reconstruction performance of PCA-reduced AE.

### 3.3. Are Low-Variance Components Merely Noise?

Having identified that VQ models suffer from severe spectral truncation compared to their continuous AE counterparts, a fundamental question arises: *Does the loss of these low-variance components actually harm performance, or are they merely noise correctly filtered out by quantization?*

To answer this, we conduct a proxy ablation study on the continuous AE baseline. We simulate the effect of spectral truncation by systematically pruning this continuous distribution. Specifically, we perform Principal Component Analysis (PCA) on the AE's latent features and project them onto subspaces spanned solely by their top-$k$ principal components. The dimension $k$ is varied to preserve different levels of cumulative spectral variance, ranging from $D_{90}$ to $D_{99}$.

As visualized in Figure 3, we observe a monotonic degradation in reconstruction fidelity as the spectral tail is removed. Notably, when the AE's spectrum is truncated to $D_{90}$—approximating the effective capacity of the standard VQ model—its performance (PSNR) deteriorates significantly, converging to the level of the VQ baseline. This convergence confirms that the low-variance components between $D_{90}$ and $D_{99}$—which are suppressed in standard VQ—are not negligible noise, but rather encode vital information for the total representational capacity. *Therefore,*

subspace. Furthermore, we observe a consistent trend in the spectral gap between AE and VQ models: the dimension count at 90% variance ($D_{90}$) remains largely comparable, whereas a substantial discrepancy emerges at the 99% level ($D_{99}$). This clarifies the nature of dimensional collapse: VQ models do not lose principal information but rather suppress low-variance components. This implies that dimensional collapse is fundamentally a spectral truncation of the tail, an inherent characteristic of vector quantization.

*mitigating spectral truncation to approximate the continuous spectral distribution is critical for enhancing the performance of VQ models.*

## 4. Method: Codebook Regularization

To mitigate the suppression of low-variance components, a direct strategy involves imposing explicit constraints on the singular values derived from the codebook's SVD. A representative approach is the Spectrum Hinge constraint, which enforces a minimum spectral magnitude by penalizing any singular value $\sigma_i$ below a threshold $\tau$:

$$\mathcal{L}_{\text{Spec}} = \sum_{i=1}^{d} \max(0, \tau - \sigma_i) \tag{5}$$

However, applying such hard constraints raises a critical issue: sensitivity. As analyzed in Sec. 5.5, this approach exhibits extreme sensitivity to its internal threshold parameter $\tau$. This sensitivity arises because artificially restricting the spectrum—for instance, by forcing all singular values to be larger than a fixed threshold—aggressively distorts the intrinsic spectral distribution of the data.

To overcome these limitations, we discard rigid constraints on singular value magnitudes and instead adopt an implicit soft constraint strategy. This term minimizes the off-diagonal correlations of the codebook covariance:

$$\mathcal{L}_{cr} = \lambda \sum_{i} \sum_{j \neq i} \text{Cov}(C)_{ij}^2 \tag{6}$$

where $\text{Cov}(C)$ denotes the covariance (or correlation) matrix calculated along the embedding dimensions of the codebook. Intuitively, by driving the off-diagonal elements of the covariance matrix towards zero, this objective effectively decorrelates the different variance component of the embeddings. This decorrelation reduces the redundancy between dimensions and encourages the model to utilize the full capacity of the embedding space, thereby implicitly revitalizing the dormant low-variance components without the need for explicit thresholding.

## 5. Experiments

### 5.1. Implementation Details

**Reconstruction Settings** To verify the universality and robustness of our proposed method, we integrate codebook regularization into three distinct VQ-GAN training paradigms: vanilla VQ-GAN (Esser et al., 2021), SimVQ (Zhu et al., 2025), and IBQ (Shi et al., 2025). Furthermore, we benchmark our approach against state-of-the-art models, including Open-MAGVIT2 (Luo et al., 2024), VQGAN-LC (Zhu et al., 2024), and LlamaGen (Sun et al., 2024). To ensure a fair comparison, we strictly adhere to

**Algorithm 1** Pseudocode of Codebook Regularization (CR)

1: **Input: Codebook** $C \in \mathbb{R}^{K \times D}$**, Loss weight** $\lambda$
    STATE **Codebook Regularization loss** $\mathcal{L}_{\textbf{CR}}$
2: {Compute the covariance/correlation matrix of the codebook}
3: $\text{Cov}(C) \leftarrow C^\top C \quad \triangleright \text{Cov(C)} \in \mathbb{R}^{D \times D}$
4: {Construct off-diagonal mask}
5: $I \leftarrow \text{eye}(D)$
6: $M \leftarrow 1 - I$
7: {Calculate the mean squared off-diagonal correlations}
8: $\mathcal{L} \leftarrow \sum(\text{Cov}(C) \odot M)^2 \quad \triangleright \odot$ denotes Hadamard product
9: $\mathcal{L} \leftarrow \mathcal{L}/(D \times (D-1))$
10: **return** $\lambda \cdot \mathcal{L}$

the original architectural configurations of the baselines; we only modify the optimization hyperparameters and the weight of our regularization term ($\lambda$). Reconstruction performance is comprehensively evaluated using rFID (Heusel et al., 2017), LPIPS (Zhang et al., 2018), PSNR, and SSIM on the validation set.

**Generation Settings** We assess the generative performance of our method on ImageNet $256 \times 256$ using an Autoregressive (AR) model. We employ the tokenizer obtained from IBQ enhanced with our codebook regularization (IBQ+$\mathcal{L}_{cr}$) at a codebook size of $K = 16,384$ for autoregressive (AR) modeling. Our AR backbone follows the same architecture and training paradigm as IBQ, which is a Llama-based model with RoPE (Su et al., 2024), SwiGLU (Shazeer, 2020), RMSNorm (Zhang et al., 2022), and AdaLN (Peebles & Xie, 2023). To verify scalability, we evaluate models across parameters ranging from 300M to 1.1B. We report Generation FID (gFID) and Inception Score (IS) (Salimans et al., 2016) to assess generation performance against other state-of-the-art methods including DiT (Peebles & Xie, 2023), LlamaGen (Sun et al., 2024), VAR (Tian et al., 2024b), MaskGIT (Chang et al., 2022), and Open-MAGVIT2 (Luo et al., 2024).

### 5.2. Visual Reconstruction Quality

We integrate our proposed codebook regularization ($\mathcal{L}_{cr}$) into a diverse set of representative VQ models. As detailed in Table 2, our evaluation spans a comprehensive range of configurations, covering different datasets (ImageNet $128 \times 128$ and $256 \times 256$), downsampling factors ($f = 8, 16$), codebook sizes ($16,384, 65,536, 262,144$), and embedding dimensions ($D = 128, 256$). Empirical results demonstrate that our simple regularization objective yields consistent improvements in reconstruction fidelity—measured by rFID, PSNR, and SSIM—across *all* settings, while preserving the original high codebook utiliza-

*Table 2.* Reconstruction performance comparison between three VQ-GAN baselines and their counterparts enhanced with codebook regularization. Gray-shaded rows denote the results of our method. The best performance on each dataset is highlighted in **bold**. The symbol † denotes results are cited directly from the paper of IBQ.

| Method | Latent dim | CB Size | Util ↑ | rFID ↓ | LPIPS ↓ | PSNR ↑ | SSIM ↑ | ER ↑ |
|---|---|---|---|---|---|---|---|---|
| *Setting: 128 × 128, Ratio 8* | | | | | | | | |
| VQGAN (Esser et al., 2021) | 128 | 16384 | 4.5% | 5.12 | 0.20 | 21.32 | 68.3 | 7.1 |
| VQGAN+$\mathcal{L}_{cr}$ | 128 | 16384 | 5.7% | 3.98 | 0.17 | 22.16 | 70.4 | 16.3 |
| VQGAN (Esser et al., 2021) | 128 | 65536 | 1.4% | 3.74 | 0.17 | 22.20 | 70.6 | 8.2 |
| VQGAN+$\mathcal{L}_{cr}$ | 128 | 65536 | 1.4% | 3.06 | 0.14 | 22.65 | 72.65 | 14.3 |
| SimVQ (Zhu et al., 2025) | 128 | 65,536 | 100.0% | 2.24 | 0.12 | 24.15 | 78.40 | 9.7 |
| **SimVQ + $\mathcal{L}_{cr}$** | 128 | 65536 | 100.0% | 1.99 | 0.11 | 24.33 | 79.5 | 14.7 |
| SimVQ (Zhu et al., 2025) | 128 | 262,144 | 100.0% | 1.99 | 0.11 | 24.68 | 80.30 | 9.3 |
| **SimVQ + $\mathcal{L}_{cr}$** | 128 | 262,144 | 100.0% | **1.55** | **0.10** | **24.99** | **81.2** | 16.4 |
| *Setting: 256 × 256, Ratio 16* | | | | | | | | |
| LlamaGen (Sun et al., 2024) | 8 | 16384 | 96.8% | 2.13 | 0.21 | 20.79 | **67.52** | 6.8 |
| VQGAN-LC† (Zhu et al., 2024) | 8 | 16384 | 99.9% | 3.01 | 0.22 | - | 56.4 | - |
| Open-MAGVIT2† (Luo et al., 2024) | 0 | 16384 | 100% | 1.58 | 0.22 | - | - | - |
| Open-MAGVIT2† (Luo et al., 2024) | 0 | 262144 | 100% | 1.17 | 0.20 | - | - | - |
| IBQ (Shi et al., 2025)† | 256 | 16384 | 95.5% | 1.53 | 0.22 | 21.10 | 57.02 | 12.4 |
| **IBQ + $\mathcal{L}_{cr}$** | 256 | 16384 | 100% | 1.32 | 0.20 | 22.53 | 59.36 | 20.3 |
| IBQ (Shi et al., 2025)† | 256 | 262,144 | 84.3% | 1.00 | 0.20 | 21.86 | 59.77 | 13.8 |
| **IBQ + $\mathcal{L}_{cr}$** | 256 | 262,144 | 100% | **0.91** | **0.19** | **22.83** | 62.13 | 21.8 |

tion rates. This robustness strongly verifies both the effectiveness and the universality of our approach. Specifically, our enhanced SimVQ and IBQ models achieve state-of-the-art performance on ImageNet $128 \times 128$ and $256 \times 256$, achieving rFIDs of 1.50 and 0.91, respectively. Regardless of the embedding dimension, the effective rank of the original VQ models hovers around 8 to 10. This validates our hypothesis that VQ codebook spaces suffer from severe dimensional collapse. In contrast, our method successfully elevates the effective rank to the range of 15–20. This significant increase demonstrates that our regularization effectively mitigates dimensional collapse, thereby fundamentally expanding the expressive capacity of the tokenizers.

### 5.3. Solving Dimension Collapse by Awakening Low-variance Components

Our method successfully resolves **dimensional collapse** by significantly increasing the effective rank across all baselines. To investigate the origin of this improvement, we analyze the metrics $D_{90}$ and $D_{99}$ between ours and baselines.

As illustrated in Table 4, the change in $D_{90}$ is negligible. Conversely, $D_{99}$ exhibits a substantial increase ranging from 4.3 to 8.9. These findings indicate that the increase in effective rank is driven by the awakening of low-variance components, which serves as a validation of our core hypothesis that the key to resolving dimensional collapse lies

in revitalizing the dormant low-variance tail. By bridging the spectral gap between the discrete and continuous representations, our approach narrows the quantization gap and achieves significant gains in representational capacity.

### 5.4. Boosting Autoregressive Generative Models

To further verify that our proposed codebook regularization yields a superior discrete representation for downstream generation, we evaluate generation performance within an autoregressive (AR) framework. Table 3 presents the class-conditional image generation results on ImageNet $256 \times 256$ based on our enhanced IBQ tokenizers. Our variant based on IBQ-XXL achieves state-of-the-art gFID results compared to all other baselines. It is worth highlighting that our method significantly outperforms the same-scale AR model trained with the original IBQ tokenizer. We attribute this performance gain to the quality of the discrete representation: our regularized tokenizer mitigates dimensional collapse, thereby encoding more comprehensive information that the AR model can leverage for high-fidelity generation.

### 5.5. Comparison with Hard Constraint Regularization

In this section, we conduct comprehensive ablation studies to validate the effectiveness of our proposed codebook regularization. Unless otherwise specified, we employ the VQGAN framework on the ImageNet $128 \times 128$ dataset, using a downsampling factor of $f = 8$, a codebook size of

*Table 3.* Quantitative comparison of autoregressive models. We report FID, IS, Precision, and Recall on ImageNet 256×256. Gray-shaded rows denote the results of our method. The best performance is highlighted in **bold**.

| Model | #Para. | FID↓ | IS↑ | Precision↑ | Recall↑ |
|---|---|---|---|---|---|
| DiT-L/2 (Peebles & Xie, 2023) | 458M | 5.02 | 167.2 | 0.75 | 0.57 |
| DiT-XL/2 (Peebles & Xie, 2023) | 675M | 2.27 | 278.2 | 0.83 | 0.57 |
| MaskGIT (Chang et al., 2022) | 227M | 6.18 | 182.1 | 0.80 | 0.51 |
| VQGAN (Esser et al., 2021) | 227M | 18.65 | 80.4 | 0.78 | 0.26 |
| VQGAN (Esser et al., 2021) | 1.4B | 15.78 | 74.3 | – | – |
| VQGAN-re (Esser et al., 2021) | 1.4B | 5.20 | 280.3 | – | – |
| VAR-d16 (Tian et al., 2024a) | 310M | 3.30 | 274.4 | 0.84 | 0.51 |
| VAR-d20 (Tian et al., 2024a) | 600M | 2.57 | **302.6** | 0.83 | 0.56 |
| Open-MAGVIT2-B (Luo et al., 2024) | 343M | 3.08 | 258.26 | 0.85 | 0.51 |
| Open-MAGVIT2-L (Luo et al., 2024) | 804M | 2.51 | 271.70 | 0.84 | 0.54 |
| LlamaGen-L (Sun et al., 2024) | 343M | 3.80 | 248.28 | 0.83 | 0.51 |
| LlamaGen-XL (Sun et al., 2024) | 775M | 3.39 | 227.08 | 0.81 | 0.54 |
| LlamaGen-XXL (Sun et al., 2024) | 1.4B | 3.09 | 253.61 | 0.83 | 0.53 |
| IBQ-B (Shi et al., 2025) | 342M | 2.88 | 254.73 | 0.84 | 0.51 |
| **IBQ-B + $\mathcal{L}_{cr}$** | 342M | 2.42 | 261.40 | 0.84 | 0.53 |
| IBQ-L (Shi et al., 2025) | 649M | 2.45 | 267.48 | 0.83 | 0.52 |
| **IBQ-L + $\mathcal{L}_{cr}$** | 649M | 2.21 | 275.12 | 0.83 | 0.55 |
| IBQ-XL (Shi et al., 2025) | 1.1B | 2.14 | 278.99 | 0.83 | 0.56 |
| **IBQ-XXL + $_{cr}$** | 1.1B | **2.02** | 284.22 | **0.85** | **0.59** |

$K = 65,536$, and an embedding dimension of $D = 128$ as the default configuration.

**Sensitivity of Hyperparameters In Hard Constraint.** To validate the superiority of our implicit soft constraint strategy, we compare it against explicit hard constraint regularization, primarily represented by the Spectrum Hinge($\mathcal{L}_{\text{Spec}}$).

A critical drawback of such hard constraints is their extreme sensitivity to internal hyperparameters—specifically the threshold $\tau$—rather than just the loss weight. As shown in Table 5, this method requires meticulous tuning. For instance, slightly increasing $\tau$ from $0.2$ to $0.3$ causes the rFID to deteriorate sharply ($4.12 \rightarrow 9.45$). We attribute this instability to the fact that hard constraints do not merely revitalize the dormant low-variance components but inadvertently distort the dominant high-variance components. This aggressive intervention corrupts the intrinsic spectral distribution of the codebook space, leading to suboptimal representation.

### 5.6. Comparison with Other Regularization Methods

To rigorously evaluate the effectiveness of our approach, we benchmark it against established regularization objectives from Self-Supervised Learning (SSL) aimed at increasing effective rank, specifically Barlow Twins (Zbontar et al., 2021) and VICReg (Bardes et al., 2022). Taking Barlow

*Table 4.* Analysis of spectral distribution changes and performance gains. We report the differential gains ($\Delta$) achieved by our method compared to the native baselines.

| Configuration | $\Delta$ **ER** ↑ | $\Delta D_{90}$ | $\Delta D_{99}$ ↑ |
|---|---|---|---|
| *Setting: $128 \times 128$, Ratio 8* | | | |
| VQGAN-16384 | +9.2 | +1.2 | +10.7 |
| VQGAN-65536 | +6.1 | +0.4 | +7.3 |
| SimVQ-65536 | +5.0 | +0.9 | +4.3 |
| SimVQ-262144 | +7.1 | +1.4 | +6.1 |
| *Setting: $256 \times 256$, Ratio 16* | | | |
| IBQ-16384 | +7.9 | +0.7 | +7.5 |
| IBQ-262144 | +8.0 | +0.3 | +8.9 |

Twins as a representative example, this method imposes constraints on the cross-correlation matrix $\text{Cov}(X)$ derived from batch features $X$:

$$\mathcal{L}_{BT} = \underbrace{\sum_i (1 - \text{Cov}(X)_{ii})^2}_{\text{Invariance Term}} + \lambda \underbrace{\sum_i \sum_{j \neq i} \text{Cov}(X)_{ij}^2}_{\text{Redundancy Reduction}} \quad (7)$$

For a comprehensive comparison, we adapt these objectives to three distinct targets within our VQ framework: the continuous encoder output ($z_e$), the batched quantized features ($z_q$), and the global codebook matrix ($C$).

*Table 5.* Sensitivity of Spectrum Hinge to the singular value threshold $\tau$. The non-monotonic fluctuation in rFID indicates high sensitivity to hyperparameter tuning.

| Threshold $\tau$ | 0.2 | 0.3 | 0.5 | 0.6 | 1.0 |
|---|---|---|---|---|---|
| ER $\uparrow$ | 18.5 | 24.2 | 10.4 | 45.8 | 32.0 |
| rFID $\downarrow$ | 4.12 | 9.45 | 3.58 | 6.45 | 7.39 |

*Table 6.* Ablation study of regularization targets: encoder output ($z_e$), quantized features ($z_q$), and global codebook ($C$). We also report commitment loss $\mathcal{L}_{commit}$ and rFID to measure the reconstruction performance. The weight of the loss are set to 0.01.

| Method | Target | ER $\uparrow$ | rFID $\downarrow$ | $\mathcal{L}_{commit} \downarrow$ |
|---|---|---|---|---|
| Barlow Twins | $z_e$ | 21.8 | 19.38 | 0.75 |
| | $z_q$ | 64.2 | 36.4 | 0.63 |
| | $C$ | 12.7 | 3.27 | 0.48 |
| VICReg | $z_e$ | 34.5 | 21.2 | 0.61 |
| | $z_q$ | 47.9 | 25.4 | 0.72 |
| | $C$ | 13.5 | 3.21 | 0.41 |
| **Ours** | $C$ | 14.3 | 3.06 | 0.39 |

**The Paradox of High Rank in Batch-wise Optimization**
As shown in Table 6, applying regularization to batch-wise features ($z_e$ or $z_q$) leads to a paradox of high rank. While these methods significantly boost the effective rank (e.g., $9.7 \rightarrow 127.5$), the reconstruction performance degrades catastrophically. We attribute this failure to the optimization mismatch induced by the Straight-Through Estimator (STE). During training, batch-wise regularization only optimizes the sampled subset of active codes. However, gradients are propagated via STE to the encoder, which updates its weights globally. This creates a conflict: the encoder's manifold shifts globally, whereas the codebook distribution—being updated only locally—fails to track this shift. This misalignment is empirically evidenced by a sharp spike in commitment loss ($\mathcal{L}_c$), indicating that the codebook fails to align with the encoder's intrinsic manifold.

Applying these objectives directly to the global codebook ($C$) resolves the optimization mismatch and improves performance. However, our proposed method still consistently outperforms these global SSL baselines. We attribute this performance gap to the nature of the constraints. Standard SSL objectives (like Barlow Twins and VICReg) impose explicit constraints on the diagonal elements of the covariance matrix (invariance term), forcing the variance of each dimension towards a fixed constant (e.g., 1). Mathematically, constraining the diagonal variance imposes a rigid restriction on the magnitude of eigenvalues, which is similar to the hard constraints discussed in Sec. 5.5.

# 6. Related Work

## 6.1. Vector Quantization Models

Vector Quantization (VQ) constitutes a foundational paradigm for mapping continuous signals into discrete tokens. The seminal VQ-VAE (van den Oord et al., 2017) established the encoder-quantizer-decoder architecture, employing a learnable codebook as the discrete representation space. To enhance reconstruction fidelity, VQGAN (Esser et al., 2021) augmented this framework with adversarial and perceptual losses. Furthermore, to expand representational capacity, methods such as RVQ (Lee et al., 2022) and MoVQ (Zheng et al., 2022) introduced residual and multi-channel quantization schemes to effectively enlarge the vocabulary size.

To mitigate codebook collapse, several strategies have been proposed, including stochastic quantization (Takida et al., 2022), codebook clustering (Zheng & Vedaldi, 2023), and rotation tricks (Fifty et al., 2024). One approach reduces the embedding dimension to minimal values (e.g., approximately 8) to encourage code usage; however, this strategy inevitably compromises the expressive capacity of the codebook. Alternatively, methods such as VQGAN-LC (Zhu et al., 2024) initialize a massive codebook (e.g., 100k) from pre-trained CLIP representations and freeze it to enforce structural priors. Nevertheless, this fixed vocabulary limits the tokenizer's adaptability and performance upper bound. Additionally, FSQ (Mentzer et al., 2024) and LFQ (Yu et al., 2023) represent a non-parametric quantization family that utilizes scalar-level values for quantized representations. While these methods achieve 100% codebook utilization, they do so at the cost of model capacity. Recent solutions, such as IBQ (Shi et al., 2025) and SimVQ (Zhu et al., 2025), have jointly optimized the entire codebook via reparameterization through a learnable transformation layer or the incorporation of index backpropagation quantization, achieving full utilization and superior reconstruction performance.

## 6.2. Dimension Collapse

Limited research has comprehensively analyzed the embedding dimension as a determinant of codebook capacity. To the best of our knowledge, Zhang et al. represent the only prior work to identify a phenomenon of dimensional collapse analogous to our findings. However, they concluded that rank regularization is ineffective in addressing this issue and instead advocated embracing the codebook's low-rank nature. In contrast, our work demonstrates that rank regularization can significantly enhance codebook performance through global optimization.

# 7. Conclusion

In this work, we have elucidated that the dimensional collapse observed in discrete representations stems from spectral truncation—the suppression of low-variance components during quantization. To mitigate this, we propose a simple yet effective codebook regularization. Our approach adopts a implicit redundancy reduction strategy that naturally revitalizes the dormant low-variance tail and aligns the spectral distribution of the discrete codebook with that of the continuous representation. Extensive experiments demonstrate that our method serves as a versatile plug-and-play module, consistently boosting effective rank, reconstruction fidelity, and generation performance.

# Acknowledgements

This research was supported by the National Natural Science Foundation of China (Grant No. 62276245).

# Impact Statement

This paper presents work whose goal is to advance the field of Machine Learning. There are many potential societal consequences of our work, none which we feel must be specifically highlighted here.

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

# A. Reconstruction Configurations

To ensure reproducibility and facilitate fair comparisons, we provide the detailed hyperparameter settings for our experiments in Table 7. We integrate our proposed Codebook Regularization ($\mathcal{L}_{cr}$) into three representative baselines: VQGAN (Esser et al., 2021), SimVQ (Zhu et al., 2025), and IBQ (Shi et al., 2025).

We strictly adhere to the original architectural configurations and training recipes of the respective baselines. For instance, we utilize the standard discriminator architecture and perceptual loss weights as reported in the original papers. The only modifications introduced are the addition of the codebook regularization term and the corresponding adjustment of its weight ($\lambda$). For the embedding initialization, we follow the default strategies: VQGAN uses a uniform initialization, while SimVQ and IBQ employ specific initialization schemes optimized for their factorization methods. Table 7 summarizes the specific values for batch size, learning rates, codebook dimensions, and loss weights used to obtain the results reported in Section 5.2.

*Table 7.* **Detailed Training Configurations.** We compare the hyperparameter settings across the three VQ frameworks enhanced with our Codebook Regularization. Note that $K$ and $D$ denote the Codebook Size and Embedding Dimension, respectively.

| Hyperparameters | VQGAN + $\mathcal{L}_{cr}$ | SimVQ + $\mathcal{L}_{cr}$ | IBQ + $\mathcal{L}_{cr}$ |
|---|---|---|---|
| Dataset | ImageNet $128 \times 128$ | ImageNet $128 \times 128$ | ImageNet $256 \times 256$ |
| Downsample Factor ($f$) | 8 | 8 | 16 |
| Codebook Size ($K$) | 16384 / 65536 | 65536 / 262144 | 16384 / 262144 |
| Embedding Dim ($D$) | 128 | 128 | 256 |
| Embedding Init. | Uniform | Gaussian | Normal |
| L2$_{Norm}$ | False | False | False |
| Num. ResBlocks | 2 | 2 | 4 |
| Channel Multipliers | 1, 2, 2, 4 | 1, 2, 2, 4 | 1, 1, 2, 2, 4 |
| Global Batch Size | 256 | 256 | 256 |
| Training Epochs | 100 | 40 | 330 Base Learning Rate |
| $1.0 \times 10^{-4}$ | $1.0 \times 10^{-4}$ | $1.0 \times 10^{-4}$ | |
| Optimizer | Adam | AdamW | AdamW |
| Commitment Weight ($\beta$) | 0.25 | 0.25 | 0.25 |
| **Codebook Reg. Weight ($\lambda$)** | **0.001** | **0.001** | **0.001** |

# B. Detailed Generation Configurations

To evaluate the generative capabilities of the discrete representations learned by our Codebook Regularization, we train Autoregressive (AR) models following the architecture and training recipe of IBQ (Shi et al., 2025). We employ a Llama-based transformer architecture equipped with Rotary Positional Embeddings (RoPE) (Su et al., 2024), SwiGLU activation (Shazeer, 2020), RMSNorm (Zhang et al., 2022), and Adaptive Layer Normalization (AdaLN) for class conditioning.

**Model Architectures and Training Hyperparameters.** We scale our AR models across three sizes: Base (B), Large (L), and Extra Large (XL), ranging from approximately 300M to 1.1B parameters. All models are trained on the ImageNet-1K dataset using the tokenizers obtained from our method (IBQ + $\mathcal{L}_{cr}$).

The detailed configurations for model architecture and optimization are summarized in Table 8. **Deviations from Original IBQ:** To adapt to our enhanced tokenizer, we introduce specific modifications to the optimization strategy. As highlighted in Table 8, we adopt a lower peak learning rate of $1.0 \times 10^{-4}$ (compared to the original $3.0 \times 10^{-4}$) and adjust the training epochs and batch sizes strategies to ensure stable convergence.

*Table 8.* **Architecture and Training Hyperparameters for Autoregressive Models. Bold values** denote our specific settings, while values in gray denote the original IBQ configuration for clear comparison.

| Configuration | IBQ-B (342M) | IBQ-L (649M) | IBQ-XL (1.1B) |
|---|---|---|---|
| *Model Architecture* | | | |
| layers ($N$) | 16 | 20 | 24 |
| heads ($H$) | 16 | 20 | 24 |
| embedding dim ($D$) | 1024 | 1280 | 1536 |
| vocab size | 16384 | 16384 | 16384 |
| block size | 256 | 256 | 256 |
| *Optimization* | | | |
| learning rate | $\mathbf{1.0 \times 10^{-4}}$ $(3.0 \times 10^{-4})$ | $\mathbf{1.0 \times 10^{-4}}$ $(3.0 \times 10^{-4})$ | $\mathbf{1.0 \times 10^{-4}}$ $(3.0 \times 10^{-4})$ |
| weight decay | 0.05 | 0.05 | 0.05 |
| warmup steps | 5000 | 5000 | 5000 |
| global batch size | $\mathbf{48 \times 64}$ $(24 \times 64)$ | $\mathbf{48 \times 64}$ $(24 \times 64)$ | $\mathbf{48 \times 64}$ $(24 \times 64)$ |
| max epochs | **250** (300) | **250** (350) | **250** (400) |
| precision | bf16-mixed | bf16-mixed | bf16-mixed |

**Sampling Hyperparameters.** During inference, we employ classifier-free guidance (CFG) to enhance the visual fidelity of generated samples. We perform a sweep over the guidance scale ($s$) and temperature ($t$) to identify the optimal settings for each model scale. We do not use Top-$k$ sampling ($k = 0$) or Top-$p$ sampling ($p = 1.0$), relying solely on temperature scaling and CFG.

Table 9 lists the specific sampling parameters. It is worth highlighting that our method achieves optimal performance with **consistently lower CFG scales** compared to the original IBQ. This indicates that our tokenizer learns a more robust and disentangled representation, requiring less artificial guidance to produce high-fidelity images.

*Table 9.* **Sampling Hyperparameters.** Comparison of inference settings. Our method requires significantly lower CFG scales (**bold**) compared to the original IBQ (gray) to achieve optimal FID, indicating superior representation quality.

| Parameter | IBQ-B | IBQ-L | IBQ-XL |
|---|---|---|---|
| CFG Scale ($s$) | **1.75** (2.25) | **1.75** (2.00) | **2.00** (2.45) |
| Temperature ($t$) | 1.10 | 1.00 | 1.15 |
| Top-$k$ | 0 | 0 | 0 |
| Top-$p$ | 1.0 | 1.0 | 1.0 |

