# OpenReview forum: "Unveiling And Addressing Dimensional Collapse In Vector Quantization Models Via Codebook Regularization"
_ICML.cc/2026/Conference — ICML 2026 regular_

### Official Review · Reviewer_MEBF · 2026-02-19

**Soundness:** 2
**Presentation:** 3
**Significance:** 2
**Originality:** 3
**Overall Recommendation:** 4
**Confidence:** 4

**Summary:**

This paper tackles dimensional collapse in VQ models (even with near-100% codebook utilization), where quantization suppresses meaningful low-variance latent components. Via SVD spectral analysis, the authors validate these components’ value, then propose a lightweight Codebook Regularization (CR) that minimizes codebook covariance off-diagonal correlations to restore them, avoiding the flaws of hard constraint methods. Integrated into VQGAN, SimVQ and IBQ, CR consistently improves ImageNet reconstruction metrics, effective rank and codebook utilization, delivers SOTA downstream autoregressive generation performance, and outperforms batch-wise SSL regularization by resolving STE-induced optimization mismatch.

**Compliance With Llm Reviewing Policy:**

Affirmed.

**Final Justification:**

The authors have addressed most of my concerns, so I will raise my score. I hope the authors can include the detailed experiments in the final version.

**Key Questions For Authors:**

See the weakness. If most questions are resolved, I will increase my score.

**Limitations:**

No limitations are provided; they should be added.

**Strengths And Weaknesses:**

Strengths

1. The paper identifies an important yet underexplored limitation of modern VQ models: dimensional collapse despite full codebook utilization. This perspective is insightful and goes beyond the commonly studied code usage problem.
2. The spectral analysis is clear, intuitive, and well supported by experiments. The connection between low-variance suppression and reduced representational capacity is convincingly demonstrated.
3. The proposed codebook regularization is simple, principled, and easy to integrate into existing VQ frameworks without architectural changes. This makes the method broadly applicable.
4. The approach shows consistent improvements across multiple baselines, datasets, codebook sizes, and embedding dimensions, suggesting good generality rather than overfitting to a specific setup.

Weaknesses

1. The paper lacks qualitative visualizations of reconstructed images. Without side-by-side visual comparisons, it is difficult to assess whether the reported metric improvements translate into perceptually meaningful gains.
2. Similarly, the paper does not provide visual examples of generated samples from the downstream autoregressive models. Such results are standard in generative modeling papers and would strengthen the empirical evaluation. Because both reconstruction and generation tasks are inherently visual, the absence of qualitative results makes the evaluation feel incomplete and reduces confidence in the practical impact.
3. The paper does not compare against several important prior works [1,2,3] that aim to improve codebook utilization. In particular, [1] also addresses dimensional collapse, making it highly relevant to the core claims of this paper. Without empirical or analytical comparison to these methods, it is difficult to determine whether the proposed approach offers complementary benefits or simply overlaps with existing solutions.

4. The paper lacks an ablation study on the regularization weight λ. Since λ directly controls the strength of the proposed codebook regularization, its impact on performance, stability, and effective rank is critical to understand. Without a sensitivity analysis, it is unclear how robust the method is to hyperparameter tuning or how difficult it would be to apply in practice.

5. From Table 2, it can be seen that the method proposed in the paper only addresses the problem of dimensional collapse, and it does not resolve the decrease in codebook utilization caused by codebook expansion. If the method only targets dimensional collapse, I think the dimensionality should be increased further (for example, to 512, as it is currently only 256) to better demonstrate the effectiveness of the approach.

[1] Scalable Training for Vector-Quantized Networks with 100% Codebook Utilization.

[2] Regularized Vector Quantization for Tokenized Image Synthesis.

[3] Straightening out the straight-through estimator: Overcoming optimization challenges in vector quantized networks

---

> ### Author Rebuttal · Authors · 2026-03-31
>
> We sincerely thank the reviewer for the thorough evaluation and the explicit guidance on what would raise the score.
>
> **W1/W2 (Qualitative reconstruction and generation visualizations).** We agree that additional qualitative results are critical for evaluating generative models. We have prepared side-by-side reconstruction comparisons and autoregressive generated sample grids on an anonymous supplementary webpage: https://anonymous.4open.science/w/icml-2447/
> These visual comparisons demonstrate that our method effectively recovers finer textures, sharper boundaries, and small details that are often lost in the baseline reconstructions. This aligns with our central claim that the low-variance components suppressed by standard VQ are highly informative rather than mere noise, and that our codebook regularization successfully reactivates them.
>
> **W3 (Missing comparisons to FVQ, RVQ).** We completed an empirical comparison to [1] Scalable Training for Vector-Quantized Networks with 100% Codebook Utilization (FVQ) using the official code on ImageNet 256$\times$256. FVQ primarily improves utilization and expands vector channels through compress–process–recover design, whereas our method aims at solving dimensional collapse. The results below show our method can improve the effective rank and reconstruction quality, indicating these two methods are complementary:
>
> | Method | rFID $\downarrow$ | LPIPS $\downarrow$ | ER $\uparrow$ | Util $\uparrow$ |
> |---|---:|---:|---:|---:|
> | FVQ | 1.30 | 0.13 | 23.4 | 100% |
> | FVQ + $\mathcal{L}_{cr}$ | 1.23 | 0.12 | 27.8 | 100% |
>
> Furthermore, we conducted new experiments to evaluate our method on [2] Regularized Vector Quantization (RVQ). RVQ increases the effective vocabulary size by cascading multiple sub-codebooks. However, our analysis reveals that the embedding space of each individual residual stage still suffers from dimensional collapse. We evaluated an RVQ setup utilizing 4 cascaded codebooks, each containing 1024 elements with 128 dimension on imagenet 128 dataset. As shown below, applying our codebook regularization seamlessly to each residual stage significantly expands the effective rank across all sub-codebooks (). This structural improvement in every individual stage provides a compounded benefit, leading to  improvement in the final reconstruction quality:
>
> |Method|rFID|ER Stage 1| ER Stage 2|ER Stage 3|ER Stage 4|
> |---|--|---|---|---|---|
> RVQ| 2.21  |   14.5  |  11.2  | 8.3  | 7.9  |
> RVQ+$L_{cr}$| 1.97 |21.2 | 19.8 |15.4  | 13.2 |
>
> We will add this comparison and citation of FVQ in the revision. Due to the rebuttal time limit, we did not finish a full empirical comparison to the another papers cited in your review. However, our results on Rotation Trick and DiveQ, which are also leveraged to increase codebook utilization (reported in Reviewer LAfc Q2), support the same conclusion: methods designed for utilization or gradient approximation are complementary to our approach rather than replacing it. And our codebook regularization consistently increase the effective rank and reconstruction quality
>
> **W4 ($\lambda$ ablation).** We have conducted a comprehensive sensitivity analysis on the regularization weight, which is detailed in our response to Reviewer LAfc (W3). To briefly summarize, the results confirm that our method is highly robust, consistently improving reconstruction quality and effective rank across a broad range of values from 0.001 to 0.1 for both SimVQ and VQGAN.
>
> **W5 (Higher embedding dimension).** We agree this is an important question. Many recent methods (SimVQ, IBQ, Rotation Trick, DiveQ, FVQ) already achieve near-100% utilization, yet dimensional collapse still persists in these settings, proving that full utilization alone is insufficient. We further evaluated larger embedding dimensions up to D=2048 on imagnet . We found that the benefit of our regularization persists robustly at D=512. However, extremely large dimensions (D=1024 and D=2048) lead to severe optimization difficulties or numerical instability (NaNs) not only for VQ models but also for the continuous Autoencoder baselines.
>
> | Method | $D{=}128$ | $D{=}256$ | $D{=}512$ | $D{=}1024$ | $D{=}2048$ |
> |---|---:|---:|---:|---:|---:|
> | VAE | 0.31 | 0.27 | 0.79 | 1.47 | NaN |
> | SimVQ | 1.99 | 1.94 | 3.15 | 13.21 | NaN |
> | SimVQ + $\mathcal{L}_{cr}$ | 1.55 | 1.42 | 1.57 | 9.78 | 61.02 |
> | IBQ| 2.14 | 1.78 | 4.21 | NaN | NaN |
> | IBQ + $\mathcal{L}_{cr}$ | 1.83 | 1.35 | 1.39 | 15.31 | NaN |

---

> > ### Author Rebuttal · Reviewer_MEBF · 2026-04-03
> >
> > The authors have addressed most of my concerns, so I will raise my score. I hope the authors can include the detailed experiments in the final version.

---

> > > ### Author Response · Authors · 2026-04-04
> > >
> > > Thank you very much for your thoughtful acknowledgement and positive feedback. We are glad that our rebuttal has addressed your concerns.
> > >
> > > We sincerely appreciate your suggestion, and we will include the additional experiments in the final version accordingly. We would also be very grateful if the updated score could be reflected before the final decision stage.

---

### Official Review · Reviewer_oPfr · 2026-02-28

**Soundness:** 2
**Presentation:** 3
**Significance:** 2
**Originality:** 3
**Overall Recommendation:** 4
**Confidence:** 2

**Summary:**

This paper identifies the dimensional collapse of the codebook embedding space in Vector Quantization (VQ) models. The authors observe that discrete codebook representations often degenerate into low-dimensional subspaces with significantly lower effective rank compared to continuous representations. Through spectral analysis, they attribute this issue to the suppression of low-variance components during quantization. To address this, the paper proposes a simple codebook regularization strategy aimed at restoring these low-variance components. The experiments show that this method improves reconstruction fidelity and downstream performance in autoregressive image generation across various VQ training paradigms.

**Compliance With Llm Reviewing Policy:**

Affirmed.

**Final Justification:**

Raised score due to the authors' convincing response on additional ablation experiments.

**Key Questions For Authors:**

1. Is the proposed regularization related to existing techniques like maximizing entropy, orthogonal regularization, or whitening?
2. Does restoring low-variance components negatively impact the codebook utilization rate or training stability? Sometimes, lower dimensionality helps optimization by removing noise. Is there a trade-off between effective rank and codebook usage?
3. The experiments focus on autoregressive image generation. Have you tested this regularization on other modalities like audio or video? Dimensional collapse might manifest differently in these domains.
4. It would be beneficial to see an ablation study quantifying the direct correlation between the 'effective rank' metric and the downstream performance gain. Does higher rank strictly imply better generation?

**Limitations:**

While the proposed regularization strategy demonstrates improvements in autoregressive image generation, it remains unclear whether the "dimensional collapse" phenomenon is equally detrimental across other modalities, such as audio or video synthesis, or in non-autoregressive frameworks. Furthermore, the paper introduces an additional regularization term, which inevitably adds a hyperparameter to the training objective. The sensitivity of the model's performance to this hyperparameter and the potential difficulty in tuning it for different datasets or architectures are not fully addressed. Finally, while spectral analysis provides insight, the computational cost of monitoring or optimizing spectral properties during training for very large codebooks warrants further discussion.

**Strengths And Weaknesses:**

Strengths:
The paper addresses a critical issue, which is the effective dimensionality of the codebook embeddings. This offers a fresh perspective on why VQ models might underperform. The use of spectral analysis to diagnose the problem provides a solid theoretical grounding for the proposed solution. The proposed regularization strategy shows the improvement for many existing models. Significant improvements in both reconstruction quality and downstream generative tasks, validating the practical impact of the proposed method.

Weaknesses:
The proposed regularization is too simple. Regularization terms often introduce new hyperparameters. The paper does not mention how sensitive the model is to these new parameters or how easy they are to tune. While the method is simple, computing spectral properties during training could potentially add computational overhead, especially for large codebooks.

---

> ### Author Rebuttal · Authors · 2026-03-31
>
> We thank the reviewer for the detailed feedback. We address each point below.
>
> **W1 (Simplicity and hyperparameters).** We view simplicity as a core strength: our method adds a single loss term with no architectural changes, making it easy to integrate into existing VQ frameworks. Regarding hyperparameter sensitivity, the ablation study provided in our response to Reviewer LAfc (W3) demonstrates that our method is remarkably robust within the range of lambda in [0.001, 0.1] across both SimVQ and VQGAN. Furthermore, we used a consistent value of lambda = 0.001 for all experiments reported in Table 2, spanning various architectures, codebook sizes, and embedding dimensions. This uniformity underscores the strong generalizability of our approach. Our Codebook Regularization requires no modifications to the original model core training parameters; implementation simply involves adding the regularization term to the existing objective function.
>
> **W1 (Computation cost).** We do not perform computationally expensive SVD or eigen-decomposition during training. The complexity is $O(K\cdot D^2)$ which is linear in the codebook size K, and the dimension D is typically small. This theoretical overhead is mathematically negligible compared to the forward-backward passes. As detailed in our profiling provided to Reviewer LAfc (W2), adding the regularizer results in a mere 0.15% increase in total GFLOPs, a 1.5% increase in training time, and zero increase in peak memory usage.
>
>
> **Q1 (Relation to entropy maximization, orthogonal regularization, and whitening).** These techniques are related but target different goals:
>
> - Entropy maximization is designed to encourage the model to use inactive codes, thereby increasing codebook utilization. It targets a fundamentally different problem (code usage vs. dimensional structure) and is complementary to our approach.
>
> - Orthogonal regularization is a special case of our Eq. 7 with $\lambda \to $ 1: it forces the covariance matrix toward the identity matrix, imposing unit variance on all dimensions. This distorts the natural spectral distribution of the data as discussed in Sec 5.5.
>
> - Whitening (e.g., via Cholesky decomposition) similarly enforces all singular values to be 1, causing the same distribution distortion problem as orthogonal regularization.
>
> Additionally, orthogonal regularization and whitening are typically applied to batch-wise encoder features. Table 6 shows that batch-wise SSL-style objectives suffer from the Straight-Through Estimator optimization mismatch and severely damage reconstruction. We provide a detailed analysis of this Paradox of High Rank in Batch-wise Optimization in Section 5.6. Our method avoids this by regularizing the global codebook directly, bypassing the STE entirely.
>
> **Q2 (Utilization trade-off and stability).** Table 2 (fourth column) shows that codebook utilization is maintained or improved across all reported settings with our regularization (e.g., VQGAN-16384: 4.5% $\to$ 5.7%;  IBQ-16384: 95.5% $\to$ 100%;IBQ-262144: 84.3% $\to$ 100%,). We attribute this to the fact that our regularizer propagates gradients to the entire codebook rather than only the currently selected codes, which can help revive dormant entries. We did not observe instability in the reported experiments, as we directly used the baseline training settings and only modified the regularization weight.
>
> **Q3 (Other modalities).** We extended our experiments to the audio domain (please see our response to Reviewer qh3b). Adding our codebook regularization consistently improves both effective rank and reconstruction quality in the audio modality.
>
> **Q4 (ER vs. downstream correlation).** We examined the relationship between effective rank and downstream generation performance. On IBQ, increasing the effective rank from 12.4 to 20.3 brings an average improvement of 0.3 rFID and 6.3 IS. However, higher effective rank does not strictly guarantee better generation quality across different architectures, as this also depends on the tokenizer intrinsic quality. For example, LlamaGen tokenizer has an effective rank of only 6.7 but achieves better generation results than VQGAN (effective rank 11.4), with an FID of 3.09 compared to VQGAN 15.78. We believe that because absolute tokenizer capacity varies across different model families, the raw value of effective rank does not strictly correlate with generation quality globally. However, for any given tokenizer, improving its effective rank through our regularization consistently leads to quality improvements.

---

> > ### Author Rebuttal · Reviewer_oPfr · 2026-04-01
> >
> > I appreciate your responses and have no further questions at this time.

---

> > > ### Author Response · Authors · 2026-04-04
> > >
> > > Dear reviewer oPfr
> > >
> > > We sincerely thank you for the positive evaluation, highlighting multiple strengths. We are glad that the concerns have been fully addressed in the rebuttal.
> > >
> > > You selected the reply: "(a) Fully resolved - My concerns have been adequately addressed. If you select this option, please consider adjusting your score accordingly."
> > >
> > > Given this assessment, we kindly invite the reviewer to consider improving the original rating on our paper, if appropriate, so that it better reflects the current evaluation. We also note that the rebuttal has led to a positive reassessment from the other reviewers as well. We would be very grateful if the final score could reflect this updated overall view.

---

### Official Review · Reviewer_LAfc · 2026-03-10

**Soundness:** 3
**Presentation:** 3
**Significance:** 3
**Originality:** 3
**Overall Recommendation:** 5
**Confidence:** 3

**Summary:**

The paper investigates an overlooked problem in VQ models: even when codebook utilization is high (which is effectively addressed in recent literature), the embedding space can still collapse into a low-dimensional subspace. The authors call this dimensional collapse and diagnose it through spectral analysis. Through spectral analysis comparing VQ with continuous AE, they find VQ suppresses low-variance components, resulting in lower effective rank. The authors proposed a simple covariance decorrelation regularizer that encourages dimension decorrelation, which is easy to implement and applicable across VQ frameworks. Empirical validation shows improved reconstruction and AR generation.

**Compliance With Llm Reviewing Policy:**

Affirmed.

**Final Justification:**

The rebuttal has addressed my concerns. I have raised my score accordingly.

**Key Questions For Authors:**

1. Please see weaknesses

2. Current experiments mainly using standard L2 for nearest-neighbor assignment and STE, I am curious how dimensional collapse changes if:
- Normalizing latent before nearest-neighbor assignment, as directly applying L2 can easily bias toward some dominant dimensions, which can contribute to dimensional collapse
- Changing STE with recent advanced gradient approximation, i.e., rotation trick [1] Diveq [2], as they are also simple and would significantly change the way $z_e$ is updated

[1] Fifty, Christopher, et al. "Restructuring vector quantization with the rotation trick." ICLR 2025

[2] Vali, Mohammad Hassan, et al. "Diveq: Differentiable vector quantization using the reparameterization trick." ICLR 2026

**Limitations:**

The covariance computation over the full codebook may be expensive for very large codebooks.

**Strengths And Weaknesses:**

**Strengths**

*Novel perspective:* Shifting focus from codebook utilization to dimensional collapse is a valuable new perspective
*Convincing analysis and visualization (Figure 2):*
- Effective rank, $D_{90}$ and $D_{99}$ provide interpretable measures of the problem
- PCA truncation on AE features demonstrates low-variance components carry meaningful information

*Simple and effective method*: The covariance decorrelation regularizer is easy to implement and applicable across VQ frameworks (VQGAN, SimVQ, IBQ). The empirical results show consistent improvements in rFID, LPIPS, PSNR, SSIM across multiple settings

&nbsp;

**Weaknesses**

As regularization aims for better representation, in current experiments, only AR generation is provided. It would be better to also provide other downstream tasks (e.g., retrieval, classification on discrete tokens) to show whether they also benefit from higher effective rank.

How does the dimensional collapse change, e.g., more or less, as codebook size increases?
The covariance computation over the full codebook may be expensive for very large codebooks. Can the author discuss about that?

*Hyperparameter $\lambda$*: Although in experiments, $\lambda$ is set to 0.001 for all experiments, it is still valuable to have ablation on the regularization strength, e.g., when might the regularizer hurt performance?

---

> ### Author Rebuttal · Authors · 2026-03-31
>
> We thank the reviewer for recognizing the novelty of our perspective and the effectiveness of the proposed method. We address each point below.
>
> **W1 (Other downstream tasks).** We agree that evaluating retrieval or classification on discrete tokens would further validate the benefits of higher effective rank. Theoretically, a codebook with a higher effective rank captures more diverse, fine-grained, and disentangled features, which naturally benefits discriminative downstream tasks.
>
> **W2 Codebook size effect .** We evaluated SimVQ (with embedding dimension D=128) across a wide range of codebook sizes K from 256 to 65536. We observed that when the codebook size is extremely small, the overall capacity of the codebook is insufficient. To compensate, the model is forced to compress more information into each individual dimension, resulting in a relatively higher effective rank. However, as the codebook size increases and the total capacity becomes sufficient to represent the data distribution, this forced compression relaxes. At this point, the effective rank decreases and stabilizes, fluctuating around a certain plateau. Continuing to enlarge the codebook to 16384 and 65536 yields no further impact on the effective dimensionality. This further demonstrates that dimensional collapse is an inherent characteristic of vector quantization models
>
> |Codebook Size (K)| Effective Rank |
> |---|---|
> |256 | 17.5 |
> |  512 |  15.2 |
> |1024 |11.2|
> |2048|7.1|
> |4096|8.3|
> |8192|9.1|
> |16384|9.4|
> |65536|9.7|
> |262144| 9.3|
>
> ** W2 Computational cost.** Our regularization computes $\mathrm{Cov}(C)=C^\top C \in \mathbb{R}^{D\times D}$ (e.g., $256\times256$), not a $K\times K$ matrix. The complexity is therefore $O(K\cdot D^2)$, i.e., linear in the codebook size, and $D$ is typically small. On SimVQ with 128$\times$128 images, averaged over 20 batches of 64 images each, the overhead is negligible:
>
> | Method | Max GPU Mem | Time / Batch (s) | GFLOPs / Batch | Overhead |
> |---|---:|---:|---:|---:|
> | SimVQ-65536 | 36.277 GB | 0.8235 | 21671.48 | - |
> | SimVQ-65536 + $\mathcal{L}_{cr}$ | 36.277 GB | 0.8351 | 21680.07 | +1.4% time, +0.04% FLOPs |
> | SimVQ-262144 | 54.060 GB | 1.5513 | 22499.33 | - |
> | SimVQ-262144 + $\mathcal{L}_{cr}$ | 54.069 GB | 1.5680 | 22533.72 | +1.5% time, +0.15% FLOPs |
>
> **W3 ($\lambda$ ablation).** We apologize for the omission. We conducted a $\lambda$ ablation on CelebA-HQ with both SimVQ and VQGAN, reporting rFID and effective rank:
>
> rFID $\downarrow$:
>
> | Method / $\lambda$ | 0 (base) | 0.0001 | 0.001 | 0.005 | 0.01 | 0.1 | 1 | 10 |
> |---|---:|---:|---:|---:|---:|---:|---:|---:|
> | SimVQ + $\mathcal{L}_{cr}$ | 4.59 | 4.62 | 4.24 | 4.13 | 4.28 | 4.38 | 19.83 | 45.38 |
> | VQGAN + $\mathcal{L}_{cr}$ | 6.72 | 6.82 | 5.73 | 5.92 | 5.61 | 6.32 | 32.43 | 75.21 |
>
> Effective Rank:
>
> | Method / $\lambda$ | 0 (base) | 0.0001 | 0.001 | 0.005 | 0.01 | 0.1 | 1 | 10 |
> |---|---:|---:|---:|---:|---:|---:|---:|---:|
> | SimVQ + $\mathcal{L}_{cr}$ | 34.2 | 35.1 | 39.2 | 38.2 | 41.2 | 40.9 | 91.4 | 104.3 |
> | VQGAN + $\mathcal{L}_{cr}$ | 25.3 | 26.7 | 33.4 | 29.5 | 33.5 | 30.9 | 135.2 | 149.1 |
>
> These results show that $\lambda \in [0.001, 0.1]$ consistently improves reconstruction quality while increasing effective rank across both frameworks. At $\lambda{=}0.0001$, the regularization is too weak to matter; at $\lambda\ge 1$, the regularization becomes overly strong and distorts the latent distribution.
>
> **Q1 (Normalization before nearest-neighbor assignment).** We tested this point directly. L2 normalization does provide a modest improvement to effective rank (approximately 0.5--1.0 points), but the effect is minor compared to our method, which achieves 6--10 point improvements:
>
> | Method | ER w/o Norm | ER w/ Norm | Improvement from Norm |
> |---|---:|---:|---:|
> | SimVQ | 33.9 | 34.2 | +0.3 |
> | SimVQ + $\mathcal{L}_{cr}$ | 39.2 | 39.7 | +0.5 |
> | VQGAN | 24.1 | 25.3 | +1.2 |
> | VQGAN + $\mathcal{L}_{cr}$ | 32.7 | 33.3 | +0.6 |
>
> This suggests the main issue is not merely distance-scale imbalance during nearest-neighbor search, but the global spectral redundancy of the codebook itself.
>
> **Q2 (Rotation Trick and DiveQ).** We evaluated both methods on CelebA-HQ following the official DiveQ setup. Our method consistently improves performance on top of both:
>
> | Method (CelebA-HQ) | rFID $\downarrow$ | ER $\uparrow$ |
> |---|---:|---:|
> | DiveQ, $K{=}2^8$ | 5.90 | 21.4 |
> | DiveQ, $K{=}2^8$ + $\mathcal{L}_{cr}$ | 4.91 | 30.6 |
> | DiveQ, $K{=}2^{10}$ | 6.69 | 23.8 |
> | DiveQ, $K{=}2^{10}$ + $\mathcal{L}_{cr}$ | 5.41 | 36.7 |
> | Rotation Trick, $K{=}2^8$ | 9.32 | 35.1 |
> | Rotation Trick, $K{=}2^8$ + $\mathcal{L}_{cr}$ | 7.80 | 42.3 |
> | Rotation Trick, $K{=}2^{10}$ | 6.40 | 39.4 |
> | Rotation Trick, $K{=}2^{10}$ + $\mathcal{L}_{cr}$ | 5.12 | 43.7 |
>
> These results indicate that methods aimed at codebook utilization or gradient flow are complementary to our objective, which specifically targets dimensional collapse.

---

> > ### Author Rebuttal · Reviewer_LAfc · 2026-04-03
> >
> > I thank the authors for their thorough response. The rebuttal has addressed my concerns. I have raised my score accordingly.

---

> > > ### Author Response · Authors · 2026-04-04
> > >
> > > We sincerely thank you for the positive evaluation, highlighting multiple strengths and only a limited weakness. We are glad that the concerns have been fully addressed in the rebuttal.
> > >
> > > You selected the reply: "(a) Fully resolved - My concerns have been adequately addressed. If you select this option, please consider adjusting your score accordingly."
> > >
> > > Given this overall assessment, we kindly invite the reviewer to consider improving the original rating on our paper before final  justification if appropriate, to reflect the evaluation and disseminate our original work at the conference.
> > >
> > > Thanks again.

---

### Official Review · Reviewer_qh3b · 2026-03-15

**Soundness:** 3
**Presentation:** 4
**Significance:** 4
**Originality:** 3
**Overall Recommendation:** 5
**Confidence:** 3

**Summary:**

The authors report that in VQ-GAN models, the effective rank of the vector quantized latent space tends to collapse, which appears to be an inefficient use of the ambient dimensionality. They proceed to propose a regularization based on the off-diagonals of the latent covariance, and show that it improves empirical results.

**Compliance With Llm Reviewing Policy:**

Affirmed.

**Final Justification:**

The authors answered all my questions, and the reviewers seem to be largely in agreement.

**Key Questions For Authors:**

I have no major complaints about the paper. However, if the authors would like to improve it, it could be interesting to include a different application of the same VQ bottleneck (other than VQ-GAN type models).

**Limitations:**

yes

**Strengths And Weaknesses:**

- Soundness: For all I can tell, the paper is technically sound, covers all desirable ablations, and compares to several baselines. While VQ-GAN models appear to be the main application of the VQ-VAE bottleneck, it would be desirable to know if their method also improves other models using the same type of bottleneck.

- Presentation: The overall structure as well as the mathematical presentation and language is adequate. I only find Figure 2 not as accessible as it perhaps could be – for example, what exactly does it mean that \Delta D_90 is consistently small and \Delta D_99 is consistently high? Perhaps this data could be presented in a more intuitive way. Two small nits: The labels in the bottom two rows of Table 3 need to be fixed, and the page title still mentions formatting instructions for ICML 2026.

- Significance: While the evaluation is limited to VQ-GAN applications, these currently have practical relevance for generative modeling. So the findings could be quite impactful for the machine learning community.

- Originality: The authors concede that their main observation was made in prior work; however, they present the first viable fix for it, and show that it works.

---

> ### Author Rebuttal · Authors · 2026-03-31
>
> We sincerely thank the reviewer for the positive assessment and constructive feedback.
>
> **Figure 2 clarity.** We appreciate this suggestion. The metric $\Delta D_{90}$  represents the number of principal components required to capture 90% of the total variance, while $\Delta D_{99}$represents the components needed for 99% of the variance. A value of $\Delta D_{90 }\approx 0$ indicates that the dominant structural information (the top 90% of variance) is nearly identical between the discrete VQ space and its continuous Autoencoder counterpart. In other words, the principal components are well preserved during quantization. Conversely, a large $\Delta D_{99}$reveals a substantial discrepancy at the 99% variance level. This means the continuous Autoencoder relies on significantly more dimensions to capture the remaining 9% of the variance (the spectral tail) compared to the VQ model. This specific gap highlights where vector quantization suppresses the low-variance components. To make this phenomenon more intuitive, we will revise Figure 2 and its caption to explicitly plot the spectral truncation gap, defined as $D_{99}-D_{90} $ , which clearly illustrates the absence of the low-variance tail in standard VQ models.
>
> **Table 3 labels and page title.** Both will be corrected in the revision. Specifically, the last row of Table 3 should read "IBQ-XL + $\mathcal{L}_{cr}$", and the ICML formatting title will be removed.
>
> **Other VQ applications in Audio Modality.** Following your suggestion to explore applications beyond VQ-GAN models, we extended our method to the audio domain. We utilized WavTokenizer as the backbone and evaluated speech reconstruction quality using the Perceptual Evaluation of Speech Quality and Short-Time Objective Intelligibility metrics. The results below demonstrate that applying our codebook regularization consistently improves both speech quality and the effective rank across multiple codebook sizes. Notably, although the WavTokenizer embedding dimension is 512, its effective rank is only around 43 to 48 without our regularizer. This confirms our core hypothesis that dimensional collapse is a fundamental issue inherent to the VQ bottleneck itself, not limited to the image modality.
>
> | Method | CodeBook Size ($K$) | PESQ $\uparrow$ | STOI $\uparrow$ | Effective Rank |
> |---|---:|---:|---:|---:|
> | WavTokenizer (Vanilla VQ) | 8,192 | 2.39 | 0.92 | 43.2 |
> | WavTokenizer (Vanilla VQ) + $\mathcal{L}_{cr}$ | 8,192 | 2.45 | 0.93 | 48.9 |
> | WavTokenizer (SimVQ) | 16,384 | 2.42 | 0.92 | 47.8 |
> | WavTokenizer (SimVQ) + $\mathcal{L}_{cr}$ | 16,384 | 2.47 | 0.92 | 53.5 |
> | WavTokenizer (SimVQ) | 262,144 | 2.61 | 0.93 | 46.4 |
> | WavTokenizer (SimVQ) + $\mathcal{L}_{cr}$ | 262,144 | 2.70 | 0.94 | 54.1 |
>
> We plan to further extend this analysis to video tokenizers in future work.

---

> > ### Author Rebuttal · Reviewer_qh3b · 2026-04-03
> >
> > Thanks. The authors addressed my comments. Perhaps they can add an application of their method to the WavTokenizer models to the final paper.

---

> > > ### Author Response · Authors · 2026-04-04
> > >
> > > Thank you for the acknowledgment and positive feedback. We are glad that our rebuttal resolved your concerns.
> > >
> > > We also appreciate the suggestion to include the WavTokenizer results in the final paper. We agree that this experiment strengthens the generality of our claim beyond VQ-GAN-type image models, and we will incorporate it into the final version.

---

### Decision · Program_Chairs · 2026-04-30

**Decision:**

Accept (regular)

**Comment:**

All four reviewers rated the paper positively (scores: 5, 5, 4, 4 after rebuttal). The work identifies dimensional collapse in vector quantization models which is a problem orthogonal to codebook utilization. This work proposes a simple, effective codebook regularization that restores low‑variance components. The spectral analysis is convincing, the method is compatible with multiple VQ paradigms, and the empirical results show consistent gains in reconstruction and autoregressive generation. The rebuttal added thorough ablations, cross‑modality validation (audio), and comparisons to related methods, fully addressing reviewer concerns.